# Deciphering clinical abbreviations with a privacy protecting machine learning system

**Alvin Rajkomar** [1,2] ✉**, Eric Loreaux**[1,2]**, Yuchen Liu**[1]**, Jonas Kemp**[1]**, Benny Li**[1]**, Ming-Jun Chen**[1]**, Yi Zhang**[1]**, Afroz Mohiuddin** [1] **& Juraj Gottweis**[1]

Physicians write clinical notes with abbreviations and shorthand that are difficult to decipher. Abbreviations can be clinical jargon (writing "HIT" for "heparin induced thrombocytopenia"), ambiguous terms that require expertise to disambiguate (using "MS" for "multiple sclerosis" or "mental status"), or domain-specific vernacular ("cb" for "complicated by"). Here we train machine learning models on public web data to decode such text by replacing abbreviations with their meanings. We report a single translation model that simultaneously detects and expands thousands of abbreviations in real clinical notes with accuracies ranging from 92.1%-97.1% on multiple external test datasets. The model equals or exceeds the performance of board-certified physicians (97.6% vs 88.7% total accuracy). Our results demonstrate a general method to contextually decipher abbreviations and shorthand that is built without any privacy-compromising data.

Patients find it difficult to understand jargon and abbreviations in the notes of their medical records[1]. Given new US legislation that mandates clinical notes be shared with all patients electronically, it is increasingly important to make this information understandable and useful to the more than 50 million patients who currently have access to their medical notes[2]. In a recent study, patient comprehension of 10 common medical abbreviations was found to be 62%, whereas expanding these abbreviations boosted comprehension to 95%[3]. Even clinicians face difficulty deciphering clinical notes due to idiosyncratic terminology and acronyms across clinical specialties and local parlance[4]. In one study, six common abbreviations in hospital discharge summaries were routinely misinterpreted by local general practitioners[5] and there is evidence that misinterpretations can cause medical harm[6]. Although a majority of physicians would prefer that discharge summaries not contain any abbreviations[7], one study found as many as 750 abbreviations contained in only 100 hospitalization discharge summaries[8]. In response, clinicians are advised to simply avoid shorthand[9,10], but this makes writing notes less efficient[11], adds administrative burden[12], and would not improve understanding of notes written in the past. There have even been calls to create autonomous systems to assist clinicians in replacing dangerous abbreviations with complete wordings[13].

Abbreviation and acronym disambiguation are part of an active area of text normalization research[14,15]. Key components include the separate tasks of detecting abbreviations in free text snippets and the expansion of those abbreviations into concepts or long forms. The number of abbreviations that prior methods have evaluated varies from 13 to 1116[15–18], with separate models typically developed for each abbreviation. Many studies include only abbreviations that are "ambiguous" (i.e., multiple long forms for the abbreviation)[18], although unambiguous abbreviations can be difficult even for physicians in other specialties to discern[19,20]. To detect abbreviations in text, prior research focuses on heuristics, such as string matching of abbreviations like "ivf," rather than machine learning[20–22]. We found 180 medical abbreviations that have dual usage as English words, such as "us" as "ultrasound" and as a pronoun, making string-matching heuristics imperfect. This also highlights that even seemingly unambiguous abbreviations can still be ambiguous in practice.

A variety of machine-learning approaches have been developed for disambiguating abbreviations in clinical text, including naive Bayes[23], support vector machines[23,24], profile-based approaches[25], algorithms based on hyperdimensional computing[26], convolutional neural networks[27], long short-term memory networks[28,29], encoder-based transformers (e.g. clinicalBERT)[18,30], latent meaning cells[31], and decoder-based transformers[32]. A recent study[18] introduced a model

---

[1]Google, Mountain View, CA, USA. [2]These authors contributed equally: Alvin Rajkomar, Eric Loreaux. ✉e-mail: alvinrajkomar@google.com

that predicts the correct expansion of a detected abbreviation from all its possible senses. The authors trained the model using reverse substitution of de-identified clinical notes, in which each long form is substituted with an appropriate abbreviation. The substituted text is then used as input and the original long form as label[17]. The investigators boosted the training data available for rare long forms by using a medical ontology to find synonymous long forms which were more common, the context of which they then re-purposed for the rarer long forms[18]. Thousands of separate models—one per abbreviation—were created. Expansions like "it" were excluded because of their ambiguous usage as English words.

Key challenges are apparent from previous work in this field. First, to the extent that clinical abbreviation disambiguation can be thought of as a form of translation, there exists no clinical corpus of original and "translated" text snippets, in which abbreviations are systematically disambiguated. While some automated machine-learning systems are able to overcome the paucity of training data by using costly or imprecise labeling techniques, a second challenge is the reliance of these systems on de-identified medical training data[33], and the concerns this type of data usage raises regarding accidental leaks[34,35]. One method to avoid the central collection of large sensitive datasets is to use federated learning[36,37], but this requires data preparation to ensure consistent data structures across sites, which does not widely exist in electronic health record systems[38]. A third challenge is the large number of separate tasks involved in comprehensive clinical abbreviation disambiguation, often requiring complex multi-model systems. Previous state-of-the-art models for abbreviation detection are trained separately from expansion models[22], and state-of-the-art models for abbreviation expansion are trained independently for each ambiguous abbreviation[18].

In this work, we investigate whether it is possible to overcome these challenges by leveraging a different training and modeling paradigm. For training, we use public web data instead of clinical text. By applying reverse substitution to web data such that it resembles clinical text with abbreviations, we can leverage an abundant source of text for our translation corpus while avoiding the use of any sensitive patient data. For modeling, we combine both detection and expansion tasks across all abbreviations into a single end-to-end translation model, simplifying serving complexity and enabling the entire clinical disambiguation procedure to be optimized with the same objective, training procedure, and decoding process. After using web data to train such a model, we then apply this model to translate real clinical text, a process referred to as transfer learning. Our approach to training and modeling introduces unique challenges of its own, such as the need for an efficient distributed algorithm for web-scale reverse substitution (WSRS), and the inherent domain shift between web data and clinical notes.

We make four key technical contributions in this paper. The first is to simplify medical abbreviation detection and expansion into a single-model, end-to-end translation system. The second is to demonstrate that such a clinical translation system can be developed solely with public web data, for which we design the web-scale reverse substitution (WSRS) algorithm[17,39]. The third is to demonstrate that a chained-inference technique called elicitive inference is sufficient to overcome the domain shift between public web data and clinical notes, resulting in the comprehensive deciphering of clinical abbreviations. The fourth and final contribution is the release of our research materials, including code to apply WSRS to the C4 web crawl dataset (C4-WSRS), code to reproduce all four clinical notes datasets and compute evaluation metrics, and the abbreviation–expansion dictionary that we manually curated for the study. Our results show that through these contributions, we are able to develop a privacy-protecting machine-learning system that achieves state-of-the-art performance on four independent clinical notes datasets, including synthetic note snippets that were written by clinicians for this study, and anonymized or de-identified clinical note snippets from three separate hospital systems.

We also evaluated the capability of four human groups to decipher clinical abbreviations: lay people with and without access to the Google search engine, medical students, and attending internal medicine physicians. We found that while access to Google can significantly boost abbreviation comprehension, there remains a gap for laypeople and that our automated system matches or exceeds physician experts in closing that gap.

## Results
### Overview

We model the abbreviation disambiguation task as a translation task, in which the snippet with abbreviations is translated to an equivalent snippet with all of the abbreviations expanded (Fig. 1A, B). This is distinct from traditional approaches in which abbreviations are detected separately and an abbreviation-specific model outputs the most likely expansion among a closed set of candidates in the dictionary. For example, consider the snippet: "This is a 45 yo m pt with chronic lbp who failed pt." The term "pt," is used in two distinct forms at different locations—the first refers to "patient" and the second to "physical therapy". In traditional approaches, a model trained to disambiguate the abbreviation "pt" would require an exogenous (e.g., manual) identification of the location of each form and two separate inference runs for each location. Our approach requires only the input snippet—identification of abbreviations is handled endogenously. Our model's self-attention mechanism is able to leverage each word's contextual representation to expand the entire snippet holistically. In this example, understanding "lbp" as "low back pain" is important to disambiguate the second instance of "pt" as physical therapy. Moreover, typically different models would be used for the other abbreviations "yo," "m," and "lbp," but our system carries out all of these detection and expansion tasks simultaneously.

To develop this system, we fine-tuned a model to take snippets containing abbreviations as input and generate the same snippet with all abbreviations expanded as output. This fine-tuning data was obtained by applying reverse substitution to public web data, resulting in artificially abbreviated snippets as model inputs and their original form as labels. The reverse substitution process (Fig. 2) uses a curated dictionary of abbreviation–expansion pairs (e.g. "af" expands to "atrial fibrillation") to first detect snippets from web pages containing clinical terms which can be abbreviated and then rewrite these snippets with the corresponding abbreviation substituted. For example, the sentence, "patients with atrial fibrillation can have chest pain," would be re-written as "pts with af can have cp." For our model, we used an encoder–decoder Text-to-Text Transfer Transformer (T5)[40] due to its natural fit for text translation tasks, its effective self-attention mechanism, and its domain-agnostic language capabilities derived from extensive self-supervised pre-training. For public web data, we used a web crawl that is analogous to Colossal Clean Crawled Corpus (C4)[40,41], but undergoes additional filtering similar to that described by Du et al.[42] Web pages were also only included if they were health-related (i.e. about a condition or symptom), and so we will refer to this web crawl as med-crawl (MC). However, due to the scale of web data, some terms with associated abbreviations occur more frequently than rarer terms (e.g. the word "patient" occurs orders of magnitude more often than "posterior tibialis"), so a simple dictionary-based substitution leads to a highly imbalanced dataset. Therefore, we created an algorithmic variant of reverse substitution amenable to distributed processing called web-scale reverse substitution (WSRS), which up-samples rare expansions, limits common ones, and is compatible with distributed computing necessary for processing web-scale data (Supplementary Fig. 1). We will refer to the fine-tuning dataset that results from applying WSRS to MC as MC-WSRS, to differentiate it from its upstream data source.

Models were evaluated on four clinical notes datasets: 302 synthetic snippets with abbreviations written by clinicians for this particular study; 21,514 anonymized snippets from real notes

**Fig. 1 | Overview of task formulation and comparison against traditional approaches. A** *Task formulation*: The input to our model is a string that may or may not contain medical abbreviations. We trained a model to output a corresponding string in which all abbreviations are simultaneously detected and expanded. If the input string does not contain an abbreviation, the model will output the original string. **B** *Traditional vs. our approach*: The traditional approach for medical abbreviation disambiguation divides the task into separate steps. First, a separate component identifies which tokens of an input string are abbreviations. Second, each abbreviation's surrounding context is processed by an abbreviation-specific model that outputs scores associated with a closed set of possible expansions of the abbreviation. This not only requires training and deploying thousands of models, but each model also cannot benefit from the inductive bias of the other models, especially as it relates to any other abbreviations that may appear in the surrounding context of a given abbreviation. Finally, each expansion must be inserted back into the original snippet, which may require additional post-processing to maintain grammatical correctness. In our approach, a single end-to-end process performs all of these steps in parallel: the detection of abbreviations, the expansion of all abbreviations such that their likelihood is maximized in a mutually consistent manner, and the generation of a grammatically consistent "decoded" piece of text.

from the University of Minnesota (referred to as CASI dataset)[43,44]; 6544 snippets created with reverse substitution from de-identified discharge summaries from the Beth Israel Deaconess Medical Center (MIMIC-III);[45] and 2888 snippets created with reverse substitution from the longitudinal records including "notes written by the doctor after a patient visit" from Massachusetts General Hospital and Brigham and Women's Hospital (i2b2-2014)[46]. The ground-truth labels (i.e. the location and expansion for each abbreviation) for the synthetic snippets were created by the writer of each snippet, whereas the labels for CASI were provided in the original dataset and the labels for MIMIC-III and i2b2-2014 were the reverse-substituted abbreviation–expansion pairs.

As our machine-learning model was trained to take an input snippet and produce an output snippet with the abbreviations expanded, it was necessary to align both snippets such that each expansion instance in the output could be attributed to the correct abbreviation instance in the input for evaluation. To achieve this, we created a token-level variant of the Needleman Wunsch global sequence alignment algorithm[47]. From those alignments we can extract abbreviation–expansion pairs and calculate four metrics:

*Detection recall (DR)* is the fraction of abbreviations that the model detects and attempts to expand. High recall means the model is correctly recognizing that the abbreviations in the snippet require expansion.

$$DR = \frac{TPD}{TPD + FND}$$

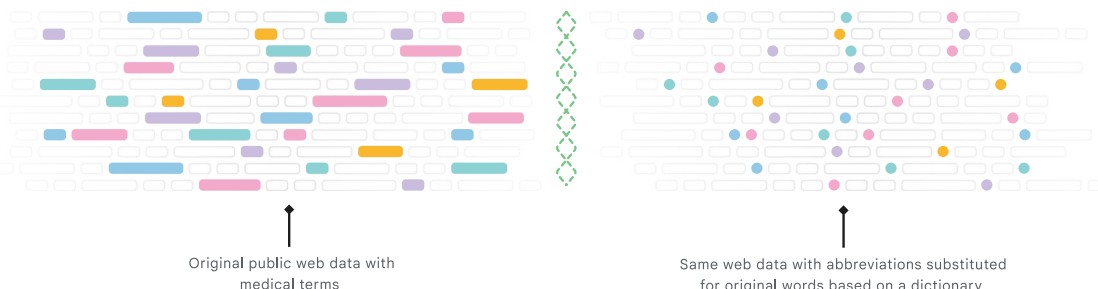

WEB SCALE REVERSE SUBSTITUTION
Full terms in public web data are converted into abbreviations using a dictionary

Original public web data with medical terms

Same web data with abbreviations substituted for original words based on a dictionary

**Fig. 2 | Overview of web-scale reverse substitution.** *Web scale reverse substitution*: Large language models are often pre-trained on public web data to carry out general self-supervised tasks. To further train (i.e. fine-tune) a model to decipher clinical abbreviations, we use the following procedure to generate an additional dataset as shown conceptually in this panel. We take public web pages and identify words or phrases that have corresponding abbreviations (the colored boxes on the left-hand side) in the dictionary released along with this manuscript. We substitute the abbreviation (the colored dots) to generate input text to train our model for the decoding task. If a word/phrase has more than one abbreviation (e.g. "atrial fibrillation" could be "af" or "afib"), we randomly pick one. Given the size of the web-corpus and the imbalance of expansions (e.g. "patient" is found orders of magnitude more often than "posterior tibialis"), a simple "find and replace" is problematic because it creates a large, imbalanced dataset. Instead, our algorithm, which we call web-scale reverse substitution, downsamples frequent expansions to derive a more balanced dataset from a web-sized corpus of thousands of expansions.

True positive detection (TPD)—model attempted to expand a word that was an abbreviation and false negative detection (FND)—model did not attempt to expand a word that was an abbreviation.

*Detection precision (DP)* refers to the fraction of the model's expansions that were attempted on actual abbreviations. High precision means the model is not modifying segments of snippets that are not abbreviations.

$$DP = \frac{TPD}{TPD + FPD}$$

False positive detection (FPD)—model attempted to expand a word that was not an abbreviation.

*Expansion accuracy (EA)* refers to the fraction of the model's abbreviation expansions that were correct. Expansions that were not an exact match but were clinically equivalent to the provided label (e.g. "CCU" could equivalently refer to "cardiac care unit" or "coronary care unit") were considered correct. An attending physician in internal medicine adjudicated whether expansions were clinically equivalent to the provided labels for each of the 302 snippets in the synthetic dataset. We release a dataset containing the sets of clinically equivalent expansion terms labeled as such for each abbreviation.

$$EA = \frac{CE}{CE + IE} = \frac{CE}{TPD}$$

Correct expansion (CE)—model's expansion for the abbreviation had the correct clinical meaning and incorrect expansion (IE)—model's expansion for the abbreviation had an incorrect clinical meaning.

The final measure was *total accuracy (TA)*, which refers to the fraction of abbreviations that were correctly detected and accurately expanded, and can be expressed as the product of detection recall and expansion accuracy.

$$TA = \frac{CE}{TPD + FND} = DR*EA$$

**Model size and inference type**
To evaluate the relationship between performance and model size, we fine-tuned four T5 models of varying sizes: T5 small (60M), T5 large (770M), T5 11B, and an 80B version of T5. The first three models had identical pre-training[40], and the 80B model was pre-trained in a similar fashion. After fine-tuning each of the models on MC-WSRS for 200,000 steps, we evaluated the models on the synthetic snippets dataset. Regarding the identically pre-trained T5 models, we observed that although model size improved expansion accuracy, the detection recall decreased, driving lower overall accuracy (Fig. 3, circle markers). Low detection recall and its decreasing trend with larger models could be caused by a number of factors. As shown in Supplementary Fig. 2, we ruled out the possibility that the number of abbreviations contained within the synthetic snippets was greater than those seen during training. We hypothesize that this resistance to expand is in part due to the fact that health-related text from the web already contains native abbreviations. These abbreviations, unaffected by reverse substitution, appear in both the model input and label during fine-tuning, teaching the model to not carry out the comprehensive expansion. Upon inspection, we found that half of the web snippets in MC-WSRS contained these native abbreviations (excluding those abbreviations which are also English words). Filtering out these examples would be undesirable, given that they represent the most medically relevant examples in the corpus.

Ultimately, we found that we could boost detection recall with inference-chaining techniques. The first technique, called iterative inference, was recently introduced in previous work and involves the use of the output of the model as the input iteratively until the model makes no modifications, at which point the process terminates[48]. The second technique, elicitive inference, is a technique we developed to take advantage of the power of beam search to induce multiple model output candidates. At the point at which the model no longer makes modifications and iterative inference terminates, a new step is introduced in which the next most likely beam for which the input is modified is chosen as input—as long as the log-likelihood score for this beam is above some fixed threshold. This continues until the input is unmodified and the next most likely beam falls below the fixed threshold (see Supplementary Fig. 3 for a diagram). The beam threshold for elicitive inference represents a hyperparameter that must be set a priori. For each model, we select the optimal threshold by applying a coarse grid search using 17 held-out synthetic snippets followed by a fine grid search using 300 held-out snippets from our fine-tuning dataset. We found that larger models required lower log-likelihood thresholds. With iterative and elicitive inference, we were able to boost detection recall and improve total accuracy statistically and substantially compared to standard inference across all model

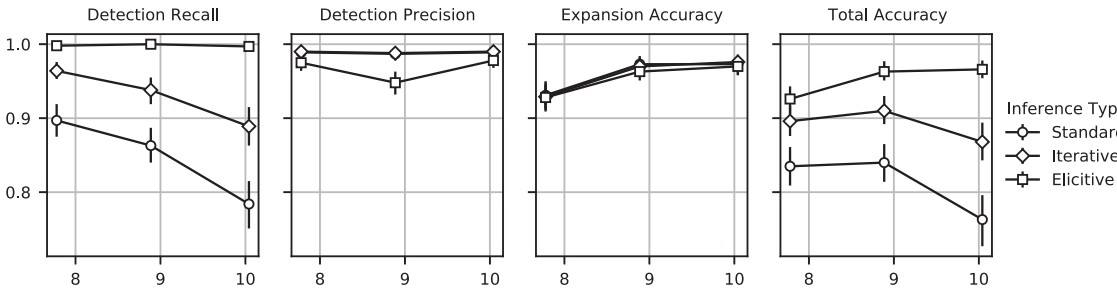

**Fig. 3 | Effect of model size and inference type on performance.** Model size and inference type influence key model metrics on the synthetic test set. Each point reflects a T5 model with identical pre-training, MC-WSRS fine-tuning, and evaluation on a synthetic dataset of medical snippets. Detection recall decreases as the model size increases. However, performance is substantially and statistically improved with inference chaining techniques such as iterative inference and elicitive inference. The inference types and model sizes do not significantly affect detection precision (percentage of the text identified as abbreviations that are actually abbreviations), and model size improves expansion accuracy (percentage of expansions with clinically equivalent meanings). Total accuracy, which we define as detection recall multiplied by expansion accuracy, is highest for the model with the most parameters with elicitive inference. $n = 400$ bootstrap samples of the 302 synthetic snippets, each of which contains a different collection of abbreviations. Point estimates from the original sample and 95% confidence intervals were calculated using reporting the 2.5 and 97.5 percentile values for each metric across the samples.

sizes (Fig. 3, diamond and square markers). With elicitive inference, larger model sizes are able to achieve higher total accuracies, with the 80B model achieving the highest accuracy of 97.0%. We use the T5 80B model combined with elicitive inference for all downstream analysis; however, the T5 11B model performed only slightly worse than this larger model. We report the raw numbers for all the models and inference techniques in Supplementary Table 1. In that table, we also report the performance of a baseline method that uses string matching to detect abbreviations and expands each abbreviation to its most commonly encountered expansion in the MC web corpus. We evaluate one version of the system in which all 180 abbreviations that can double as English words are left unexpanded, and one in which they are always expanded, highlighting the unavoidable recall/precision tradeoffs inherent in applying deterministic find-and-replace systems to such abbreviations.

## Performance on clinical text

Table 1 reports the results of the T5 80B model using elicitive inference on four test sets (described in the "Methods" section). For the synthetic snippets, CASI, MIMIC-III, and i2b2-2014 test sets, the respective detection recall was 99.1%, 96.8%, 99.5%, and 99.7%, expansion accuracy was 97.9%, 95.1%, 96.1%, and 96.7%, and total accuracy was 97.0%, 92.1%, 95.7%, and 96.5%. Results are also separated into abbreviations that are ambiguous (meaning the abbreviation has more than one valid expansion) versus those that are unambiguous. While detection recall was comparable between ambiguous and unambiguous abbreviations, expansion accuracy was higher for the latter group, surpassing 99% in all datasets. For MIMIC-III, we also stratified performance based on the rarity of the abbreviations, which we quantified using the entirety of the original discharge summary dataset (Supplementary Table 2). We found that the system's performance is independent of the rarity of abbreviations.

For all but the synthetic dataset, in which outputs were exhaustively reviewed by a physician for clinical equivalency, the expansion accuracy is a lower bound estimate, as clinically equivalent expansions not previously observed may not have been labeled as such. In addition, while the detection precision for the synthetic snippets was 99.3%, we do not report this metric for either the CASI, MIMIC-III, or i2b2-2014 test sets because of the high frequency of abbreviations which may require expansion but have not been labeled as such. This is because each CASI snippet only contains a label for a single target abbreviation, and the MIMIC-III and i2b2-2014 snippets only contain labels for those abbreviations synthetically injected by RS. We also

evaluate the model's performance on native abbreviations that were not synthetically injected (i.e. originally entered by the noted author) in MIMIC-III and i2b2. We randomly sampled 154 unique, native abbreviation–expansion pairs each from both the MIMIC-III and i2b2-2014 datasets. An attending physician in internal medicine graded each abbreviation–expansion pair in the context of the snippet. In MIMIC-III the expansion accuracy was 95.5% (147/154) and in i2b2-2014 it was 97.4% (150/154), which was consistent with the performance on abbreviations introduced by reverse substitution.

## Abbreviations that are also English words

One natural language understanding task that is implicitly performed by the model is the detection of abbreviations among the natural clinical text. This task is nontrivial because words like "it" are both English words and abbreviations. While the metrics above include such abbreviations, we sought to evaluate them separately as well. To this end, we identified 180 abbreviations in our dictionary that are also English words. Of these, 25% (44/180) were found in the synthetic snippets across 550 instances, in which 6.9% (38/550) were used as abbreviations and the rest were used as English words. Of the English word instances, 99.2% (508/512) were correctly left unexpanded, and the four "errors" included converting "post" to "after" ($N = 1$), and "per" to "by" ($N = 3$), which are meaning-preserving modifications. Of the words intended as abbreviations, all 38 resulted in accurate detection and expansion, demonstrating that the model is capable of accurately distinguishing between English word usage and abbreviation usage.

## Comparison with human performance

Given that we have reformulated abbreviation–expansion as a type of translation, we sought to compare the translation capability of our model with that of human translators across different skill levels. We randomly selected 30 snippets from the synthetic snippets dataset with at least three abbreviations each and had individuals in four groups carry out translation: three lay people doing the task themselves, three different lay people doing the task with the aid of the Google Search engine, three US medical students, and three US physicians who were board certified in internal medicine. The latter two groups were not allowed to use Google searches to simulate a time-pressured clinical environment. All groups were instructed to not expand abbreviations they were not reasonably confident in.

The laypeople without access to Google achieved a mean total accuracy of 28.6%, with a low performance largely driven by a lack of attempts to expand abbreviations (i.e. mean detection recall of 34.9%)

**Table 1 | Results on external test sets**

| | Ambiguous | | | Unambiguous | | | All | | |
|---|---|---|---|---|---|---|---|---|---|
| | DR | EA | TA | DR | EA | TA | DR | EA | TA |
| Synthetic | 0.990 (0.981, 0.997) | 0.971 (0.957, 0.983) | 0.961 (0.946, 0.975) | 0.996 (0.983, 1.000) | 1.000 (1.000, 1.000) | 0.996 (0.986, 1.000) | 0.991 (0.984, 0.998) | 0.979 (0.969, 0.987) | 0.970 (0.959, 0.981) |
| CASI | 0.967 (0.964, 0.969) | 0.949 (0.946, 0.952) | 0.917 (0.914, 0.921) | 0.995 (0.990, 0.999) | 0.999 (0.997, 1.000) | 0.994 (0.989, 0.998) | 0.968 (0.966, 0.971) | 0.951 (0.949, 0.954) | 0.921 (0.918, 0.925) |
| MIMIC-III | 0.996 (0.995, 0.997) | 0.944 (0.940, 0.948) | 0.940 (0.936, 0.945) | 0.993 (0.991, 0.995) | 0.992 (0.989, 0.994) | 0.985 (0.982, 0.988) | 0.995 (0.994, 0.996) | 0.961 (0.959, 0.964) | 0.957 (0.953, 0.960) |
| i2b2-2014 | 0.996 (0.994, 0.998) | 0.954 (0.947, 0.960) | 0.950 (0.944, 0.958) | 0.999 (0.998, 1.000) | 0.996 (0.992, 0.999) | 0.995 (0.991, 0.998) | 0.997 (0.996, 0.999) | 0.967 (0.963, 0.972) | 0.965 (0.960, 0.970) |

This table contains metrics for the T5 80B model fine-tuned on MC-WSRS using elicitive inference on the four external test sets. Confidence intervals are derived from 400 bootstrap samples of the snippets, each of which contains a different collection of abbreviations. 95% confidence intervals are calculated using the 2.5 and 97.5 percentile values for each metric across the 400 samples.

(Table 2). Access to Google substantially improved detection recall to a mean of 83%, driving mean total accuracy up to 74.5%. Both medical students and physicians performed with mean total accuracies of 88.7%. The T5 80B model combined with elicitive inference demonstrated the highest total accuracy (97.6%). Part of this performance discrepancy was due to the physicians not expanding abbreviations that are commonly used as abbreviations in their clinical workflow (e.g. not expanding "cm" to "centimeter"); however these expansions can be important to non-English-speaking patients who rely on translation services which do not work with abbreviations. We show examples of various human-made mistakes alongside T5 80B model outputs in Supplementary Table 3.

### Data quality

To understand the impact of the quality of the web data used for reverse substitution, we also fine-tuned all three T5 models on a dataset derived from the publicly available Colossal Clean Crawled Corpus (C4)[40,41], which contains clean English text scraped from the web (version 3.0.1 in Tensorflow Datasets). We refer to this fine-tuning dataset as C4-WSRS and make it available through Tensorflow Datasets. We repeated model training with identical hyperparameters and evaluated the models on the synthetic snippet dataset (Supplementary Table 4). Although there was a drop in detection recall relative to models fine-tuned on MC-WSRS, this gap was completely eliminated by elicitive inference. Detection precision was approximately the same. Expansion accuracy was slightly lower than MC-WSRS models of the same size. We evaluated the T5 11B on the three additional test datasets (Supplementary Table 5) and found that these similarities held consistently for all other test datasets except CASI, for which the C4-WSRS model exhibited a significantly lower detection recall. This drop was largely driven by a resistance to expanding common bigrams containing abbreviations, such as medication names (tylenol pm, bactrim ss) and other clinical terms (free t4, av block). We attribute this accuracy discrepancy in part to the fact that C4-WSRS does not contain the diversity or quantity of medical long forms present in MC-WSRS (Supplementary Fig. 6), demonstrating the value of enriching web crawls with medical content.

### Discussion

We report a state-of-the-art machine learning system that can achieve clinician-level performance on the task of translating abbreviated clinical text into fully expanded text. Critically, the system was built in a privacy-protecting way using only public web data. Our machine learning model can detect thousands of abbreviations across specialties in natural clinical text and simultaneously expand them with an accuracy that matches or exceeds physicians. High accuracy was maintained across multiple real clinical notes datasets from independent health systems, spanning both inpatient and outpatient visits. We provide numerous examples of the T5 80B model inputs and outputs across a variety of patient and clinical scenarios in Table 3, demonstrating the extensive natural language capabilities of the model. For example, the phrase "nr, nr" as part of a cardiovascular exam is correctly expanded to "normal rate, normal rhythm," which requires contextual understanding to expand a single abbreviation into two long forms successively. Importantly, the model can disambiguate the same abbreviation used multiple times with different intended expansions in a single sentence. This suggests that each expansion is performed with an implicit understanding of concurrent expansions (e.g. "pt" is likely to mean "physical therapy" in the context of "lbp," which means "low back pain").

Our data processing scheme overcomes two challenges highlighted by previous work, namely the lack of large corpora of original and translated text snippets, and the need to use sensitive clinical notes data for model development. Our reverse substitution method, which we call web-scale reverse substitution (WSRS), enables large-

**Table 2 | Performance of various human groups and models on the random selection of synthetic test set examples**

|  | DR | DP | EA | TA |
|---|---|---|---|---|
| Layperson | 0.349 (0.269,0.429) | 0.987 (0.973,1.000) | 0.828 (0.697,0.959) | 0.286 (0.227,0.345) |
| Layperson with Google | 0.829 (0.787,0.872) | 0.988 (0.982,0.993) | 0.900 (0.827,0.973) | 0.745 (0.701,0.790) |
| Medical Student | 0.913 (0.904,0.922) | 0.989 (0.967,1.011) | 0.971 (0.966,0.977) | 0.887 (0.882,0.892) |
| Attending Physician | 0.924 (0.903,0.944) | 0.975 (0.966,0.984) | 0.960 (0.940,0.981) | 0.887 (0.856,0.918) |
| T5 80B + elic. | 0.984 (0.961, 1.000) | 1.000 (1.000, 1.000) | 0.992 (0.976, 1.000) | 0.976 (0.945, 1.000) |

This table contains the respective performances of each group of human evaluators on the 30-snippet sample of the synthetic test set, and the T5 80B model combined with elicitive inference on the same sample. Metrics for the human evaluator groups are presented as mean values with standard error of the mean using the $n = 3$ human evaluators within each group. For the model metrics, confidence intervals are derived from 400 bootstrap samples of the snippets, each of which contains a different collection of abbreviations. 95% confidence intervals are calculated using the 2.5 and 97.5 percentile values for each metric across the 400 samples.

**Table 3 | Illustrative demonstration of model capabilities**

| Original string | T5 80B + elic. | Comment |
|---|---|---|
| 45 ftm pw f cp and sob | 45 trans man presenting with fever chest pain and shortness of breath | The same abbreviation "ftm" is grammatically expanded as a noun or adjective depending on the context, which may also have other abbreviations. |
| 45 ftm pt pw f cp and sob | 45 female to male patients presenting with fever chest pain and shortness of breath | |
| The cv exam has a nr, nr, nl s1 s2, no mrg | The cardiovascular exam has a normal rate, normal rhythm, normal first heart sound, and second heart sound, no murmurs rubs, or gallops | The model expands "nr" into two separate, correct meanings, which would be difficult to hand-engineer. The model inserts "and" to make the sentence grammatical. |
| dfe reveals macular drusen, ou | Dilated fundus examination reveals macular drusen, both eyes | Abbreviations in fields, such as ophthalmology may be unfamiliar to other clinicians |
| 45 yo m pt undergoing pt for lbp was referred for bruising and elevated pt and ptt | 45-year-old male patient undergoing physical therapy for low back pain was referred for bruising and elevated prothrombin time and partial thromboplastin time | "Pt" is correctly expanded into three separate meetings even in the presence of multiple other abbreviations |
| ms w is a 45 yo w with ms was given ms for ms pain in l leg and was found to have altered ms in the ed | ms. w is a 45-year-old woman with multiple sclerosis who was given morphine sulfate for musculoskeletal pain in her left leg and was found to have altered mental status in the emergency department | This example has 5 different long forms of "ms," and the model can successfully choose not to expand "ms" as shown in the first instance. The abbreviation "w" is also correctly ignored in the first instance and expanded to "woman" in the second. |
| The importance of electrocardiogram (ecg) in the presence of cardiac symptoms cannot be overstated | The importance of electrocardiogram (ecg) in the presence of cardiac symptoms cannot be overstated | The model recognizes that the abbreviation is being used as a didactic tool here and leaves it unexpanded, preserving the original intent of the statement |
| heent: perl, clear op, cv: rrr nl s1 s2 no mrg. abd:nabs, soft nt nd, ext: wwp, no edema, neuro: alert and oriented, psych: mildly restricted affect, labs reviewed and notable for bmp with cr of 1.7, hgb 13.0, inr 1.3, cxr without infiltrate, and ekg nsr at 61 bpm and one pvc | Head eyes ears nose throat: pupils equally round and reactive to light, clear oropharynx, cardiovascular: regular rate and rhythm normal s1 s2 no murmurs rubs or gallops. abdomen:normoactive bowel sounds, soft non-tender nondistended, extremities: warm and well per-fused, no edema, neurologic: alert and oriented, psychiatric: mildly restricted affect, labs reviewed and notable for basic metabolic panel with creatinine of 1.7, hemoglobin 13.0, international normalized ratio 1.3, chest radiograph without infiltrate, and electrocardiogram normal sinus rhythm at 61 beats per minute and one premature ventricular contraction | The model has the capacity to expand a large number of abbreviations simultaneously |

This table contains example outputs of the T5 80B model fine-tuned on MC-WSRS combined with elicitive inference, which illustrates the extensive abbreviation–expansion capabilities of this model across a variety of challenging clinical scenarios.

scale synthetic generation of abbreviated public web text by substituting native long forms on the web with their corresponding abbreviations. Although reverse substitution applied to clinical note data has been investigated before[17,39], our method is compatible with distributed computing, allowing us to process skewed web-scale data and achieve balanced sampling across a large dictionary. Another challenge we address is the large number of separate tasks required for end-to-end clinical abbreviation disambiguation, which we overcome by combining the detection and expansion of thousands of abbreviations into a single end-to-end translation model. Using a single model reduces the complexity of developing such a system in production, and enables a single training dataset and loss function to be used to develop all necessary capabilities. We unify these tasks by training a large sequence-to-sequence transformer model to translate an abbreviated snippet such that all abbreviations are simultaneously detected and expanded. We show that this system works well for abbreviations of all levels of rarity, and for both ambiguous and unambiguous abbreviations. Notably, the model is capable of complex natural language understanding challenges, such as correctly choosing to expand English words which can also double as abbreviations (e.g. the string "it" meaning the word "it" or standing for "iliotibial") based on context—a task that was previously excluded from research on disambiguating abbreviations.

Our use of public web data and end-to-end translation models introduced a novel set of challenges that initially prevented us from achieving high performance on clinical notes. We found that training a language model on a web-scale reverse substitution corpus alone and then applying the model to real clinical notes (transfer learning[49]) does

not work well out of the box. We demonstrate this is true even for very large models, suggesting that the fine-tuning data generated by WSRS represents a fundamentally different task than that of comprehensively translating real clinical text. We show that we are able to close this generalizability gap using an inference-chaining technique we call elicitive inference, in which the multiple output sequences a beam search can be used to elicit additional expansions. We emphasize that inference-chaining techniques improve model performance without updating any model parameters or using additional input data in the target domain (e.g. few-shot learning or additional fine-tuning[50]). This study demonstrates the feasibility of protecting patient privacy by using entirely public web data to develop systems that achieve expert-level performance on clinical natural language understanding tasks.

We can compare our end-to-end approach to prior state-of-the-art research. For abbreviation detection, Wu et al. reported results for 332 unique abbreviations within a dataset derived from the same corpus that was used for training and 1016 unique abbreviations within an external dataset[20]. Their best-reported detection recall for these two datasets was 71.0% and 26.9%, respectively, with corresponding best precision of 91.0% and 50.3%. Our T5 80B model achieved detection recall ranging from 96.8% to 99.7% on four clinical text datasets with the number of unique abbreviations ranging from 64 to 2485. On the only dataset for which precision could be reliably computed (synthetic snippets), our model achieves a precision of 99.3%. For abbreviation disambiguation, Skreta et al. trained a model on MIMIC and tested it on a held-out test set of MIMIC and also CASI, which represented an external test set (i.e., CASI data was not used for training)[18]. For the MIMIC internal test set, the investigators report a micro-accuracy of 93.5% across 1116 abbreviations compared to our T5 80B model's performance on MIMIC as an external test set (i.e., MIMIC data was not used for training) of 95.7% micro-accuracy across 2485 abbreviations. On CASI, the investigators report a micro-accuracy of 84.1% compared to 92.1% for our T5 80B model. Recently, Agrawal et al.[32] demonstrated that large language models can be used in a zero-shot manner to outperform classical disambiguation systems. Our T5 80B model achieves higher accuracy across more abbreviations in both CASI (92.1% on 64 abbreviations vs. 90% on 41) and MIMIC (95.7% on 2485 abbreviations vs. 79% on 41), demonstrating that fine-tuning can lead to both accurate and scalable results across datasets and abbreviations.

It is worth noting that a direct comparison between our end-to-end system and previous disambiguation approaches is not possible, since our system must detect an abbreviation in addition to expanding it, whereas previous approaches are applied to an exogenously identified abbreviation. This means that for each expansion, our system is evaluated on both detection and expansion, whereas previous approaches are solely evaluated on expansion. In addition, the abbreviation–expansion pairs evaluated depend on the dictionary used to generate the examples, and the pairs we evaluate our system on are more numerous than those evaluated in previous work.

There are several limitations of this study. First, the elicitive inference used to preserve high abbreviation detection recall introduces additional computational cost in the form of multiple sequential rounds of model inference (Supplementary Fig. 4). In future work, we hope to improve the model's ability to detect abbreviations in a single round of inference. Second, we do not compare our models to other high-performing language models in the literature, such as decoder-only models[42,51]. Our goal in this work was to leverage an encoder–decoder T5, a canonical sequence-to-sequence large language model family which has demonstrated state-of-the-art performance on a number of NLP/NLU tasks, and with which we demonstrate expert-level performance on this task. We also assessed how to size/capacity within this model family related to performance. Future work

could explore comparative performance among different large language models.

Third, human performance on the task is likely to vary based on multiple factors such as general literacy, health literacy, and for physicians, their specialty. The laypeople in the human evaluation study may not be representative of the level of education or healthcare knowledge present in the general population; however, their familiarity with internet search engines likely provides a conservative estimate of how many abbreviations can be understood with online searching. Moreover, it is possible that clinicians in different specialties may have better performance on certain types of abbreviations, such as ophthalmologists encountering eye-related abbreviations. However, the performance of internal medicine physicians given their broad training across many conditions likely provides an optimistic clinical performance assessment.

Fourth, there are risks introduced whenever a system generates output sequences in an unconstrained way. Previous methods to disambiguate abbreviations are not susceptible to these risks, since models are only applied to select abbreviations, and expansions are chosen among fixed sets. Encouragingly, we found that in practice, our system rarely expanded or modified words that were not abbreviations (detection precision of nearly 100%), and for words intended as abbreviations, all of the system's expansions came from the abbreviation dictionary used to generate the fine-tuning data. This aligns with previous research showing that model fine-tuning establishes effective constraints on large language model output[40].

Additionally, clinical terms, even in their expanded form, can still be unfamiliar to other clinicians or confusing to patients, so accurate expansion does not guarantee clinical comprehension. We acknowledge that further work is necessary to understand and improve comprehension of clinical notes across a variety of audiences.

Finally, the clinical effect of model errors in expanding abbreviations is unknown. Although we demonstrate that the model can expand abbreviations for rare diseases and expressions (Supplementary Table 6), there is no dataset we are aware of that has de-identified notes of a wide sample of rare diseases that can be used to quantify the error rate in these cases. It is unclear whether not attempting to expand rare instances of abbreviations is preferable to expanding them with a given error rate.

From a machine learning fairness angle, we point out that some abbreviations might reflect terminology which may be considered inappropriate in some contexts, such as gendered terms in the US or in the UK hospitals where openly transgender or nonbinary persons give birth[52]. Our current focus was to faithfully translate the abbreviations in the text as written, but we acknowledge that some notes themselves contain language with abbreviations that are potentially insensitive, inappropriate[53], or offensive[54,55]. Additional efforts are needed to address bias in the clinical notes.

In summary, we demonstrate a successful general method to contextually expand all clinical abbreviations in an end-to-end fashion that is built without any sensitive data (i.e., privacy-protecting). We show that our method is comparable to expert human performance. We found that lay people cannot understand more than half of the abbreviations in clinical text, and even after searching the internet for meaning, they still found 1 in 5 indiscernible. These results underscore the need for automated systems to aid in the deciphering of these abbreviations. As patients increasingly gain access to their medical records, decoding the abbreviations and shorthand in notes has the potential to enable better patient understanding by making it possible to search for information about concepts mentioned as otherwise inscrutable abbreviations. Future work is required to solve the last-mile problem[56] to not only help patients interpret their own medical records in a seamless experience[16] but also improve comprehension for those with lower health literacy, which may require not only deciphering abbreviations but also appropriately simplifying terms.

# Methods

## Overview

Our machine-learning system consists of three separate components: fine-tuning dataset generation, model fine-tuning, and model inference. For fine-tuning dataset generation, we use a distributed processing algorithm we call web-scale reverse substitution (WSRS) to substitute long forms with their abbreviations in snippets from a large web corpus. For model fine-tuning, we train large transformer models to detect and expand abbreviations by using the abbreviated snippet as model input and the original snippet containing the long forms as the label. For model inference, we chain multiple rounds of inference using a technique we call elicitive inference, in which the model's output is fed again as input to elicit further model expansions. An overview of this three-component system is shown in Supplementary Fig. 5.

The Advarra Institutional Review Board determined that this research was exempt from review and the requirement for informed consent because it did not involve individually identifiable data and thus did not qualify as human subjects research according to 45 CFR 46.

## Fine-tuning dataset generation with WSRS

Since the data in the pre-training corpus was from websites that largely did not specifically contain examples of abbreviations in clinical text, we algorithmically created example snippets from the web corpus that contained clinical terms. At a high level, we used the dictionary (described below) that contained expansions (also referred to as long forms or senses) and their abbreviations (e.g. atrial fibrillation: AF), to systematically replace expansion phrases from the public web with their abbreviations. Specifically, for each webpage used for fine-tuning, we divided the text of the page into variable-length snippets of between one and three sentences. We sequentially sampled sentences and replaced text that contained a long form in the dictionary with its abbreviation. If the long form contained more than one abbreviation ("atrial fibrillation" can be abbreviated as "af" or "afib"), we randomly selected one per sentence. All terms within the snippet were eligible to be replaced with their abbreviations. This process is referred to as "reverse substitution."

There are two key obstacles to traditional reverse substitution[17] using large-scale web data. First, frequent term replacements such as "pt" for "patient" are created at rates that are orders of magnitude higher than rare ones such as "cdgs" for "carbohydrate-deficient glycoprotein syndrome." Second, if the model only sees examples of snippets containing abbreviated forms (e.g. "us" for "ultrasound" or "as" for "aortic stenosis") then it would not learn to accommodate snippets that do not require expansions. We, therefore, created the following modified reverse substitution protocol to upsample rarer abbreviations and to create "negative" examples without abbreviations. We refer to this as web-scale reverse substitution (WSRS) and provide pseudocode for this algorithm in Supplementary Algorithm 1.

As the processing of webpages was done in parallel, pages were randomly distributed into shards for processing[57]. We filter out snippets >1024 characters in length. For each snippet, we iterate over each long form, and for the ith long form, calculate $p^i_{keep\_rs} = 1/(n^i_{rs} + 1)^{\alpha_{rs}}$, where $n^i_{rs}$ is incremented each time that long-form is reverse-substituted in that shard, and $\alpha_{rs}$ is a hyper-parameter. For each snippet, we used the maximum probability $\max(p^0_{keep\_rs}, p^1_{keep\_rs}, ...., p^L_{keep\_rs})$ of all L long forms to decide whether to keep the snippet for reverse substitution; this step limits the number of times a frequent abbreviation-expansion pair will occur in the fine-tuning set while upsampling snippets containing rare pairs. If a snippet was kept, we gave each long form a 95% probability of being substituted with an abbreviation. We filter out snippets whose abbreviated forms are <3 tokens in length. For snippets that are not sampled for a reverse substitution or result in no substitutions even after sampling, we use a probability $p_{keep\_no\_rs}$ to determine whether that snippet is still included in the fine-tuning set. This ensures that the model is exposed to snippets without any required modifications. We calculated $p_{keep\_no\_rs} = 1/(n_{no\_rs} + 1)^{\alpha_{no\_rs}}$, where $n_{no\_rs}$ is incremented each time a snippet is sampled without reverse substitution, and $\alpha_{no\_rs}$ is a hyper-parameter. Once all shards were completed, we grouped snippet examples from all shards by their rarest reverse substitution. We then randomly sampled a max threshold of N snippets from each group.

For both MC-WSRS and C4-WSRS, we set $\alpha_{rs}$ to 1.0 and $\alpha_{no\_rs}$ to 1.5 (tuned to keep the percent of snippets without reverse substitution to be less than 5% of the total). Due to differences in libraries and frameworks between the two WSRS implementations, MC-WSRS increments $n_{no\_rs}$ and $n^i_{rs}$ on a worker level, whereas C4-WSRS increments these values on a document level. The former results in more significant downsampling. For both MC-WSRS and C4-WSRS, we set the max threshold N to 1000. The result is a balanced fine-tuning dataset of snippet examples with sizes of ~5 million and ~3 million for MC-WSRS and C4-WSRS, respectively.

## Model fine-tuning

The models we use in this study are encoder-decoder Text-to-Text Transfer Transformers (T5), which are designed to take text as input and generate text output[40]. We experimented with T5 small (60M), T5 large (770M), T5 11B, and a T5 80B variant. We use the T5 80B for our main results. Each model was pre-trained on a web corpus using masked language modeling (MLM) loss[40]. We used the same vocabulary as MT5 with 250,000 wordpieces[40] covering 101 languages with byte fallback. The vocabulary was not domain-specific and was not tuned for the abbreviation task. Once pre-trained, we fine-tuned the model for the clinical abbreviation deciphering task. The training examples consisted of the modified snippet with abbreviations as the input and the original snippet with the long forms as the target. We prefixed all inputs with the string "expand abbreviations:". The training example then had the form: "expand abbreviations: <input snippet with abbreviations>" and the target/label was "<input snippet with the abbreviations expanded>". The masked language modeling loss function was used for fine-tuning. We use the Adafactor[58] optimizer with a decay rate of 0.8; a learning rate of 1e−3 with a linear warmup of 1000 steps; a batch size of 64; and a dropout rate of 0.1. All T5 models were fine-tuned for 200,000 steps.

## Model Inference

Model inference is conducted with a beam-search size of 2. Three different model inference procedures are reported in this study (see Supplementary Fig. 3 for a diagram).

- *Standard inference*: the input text is fed to the model and the output is served.
- *Iterative inference*: the input text is fed to the model. If the model's output is different from the input text, then the output is fed again as input to the model. Once the model's output is unchanged from the input text, it is served[48].
- *Elicitive inference*: the input text is fed to the model with beam search enabled. Similar to iterative inference, the model's output is fed again as input until the top-scoring beam is unchanged from the input. At this point, if the model's second-highest beam has a log-likelihood score above some fixed threshold, we treat this beam as the new input and continue the process. Once the model's top-scoring beam is unchanged from the input and the second beam has a log-likelihood below the fixed threshold, the output is served.

Among the three inference methods, elicitive inference is the only method that depends on a hyperparameter of its own: the log-likelihood threshold below which the model's second-highest beam must fall to trigger the termination of inference. For each model, we

select the optimal threshold by applying a course grid search using 17 held-out synthetic snippets followed by a fine grid search using 300 held-out snippets from our fine-tuning dataset. The optimal threshold was found to be $10^{-1.5}$ for the T5 small and T5 large models and $10^{-2.8}$ for the T5 11B model.

## Clinical text datasets

**Synthetic snippets**. We generated 302 snippets of clinical text containing abbreviations by asking senior medical students, residents, and attending physicians board certified in internal medicine to generate sentences that contained medical abbreviations that we randomly sampled from our dictionary; these snippets are released publicly as part of this manuscript. Since many abbreviations are ambiguous, we generated snippets for each distinct clinical meaning expansion (e.g. "af" had a snippet that expanded to "atrial fibrillation" and another to "afebrile"). Clinicians could use other abbreviations not in the given list as appropriate to make realistic synthetic text. These clinicians also created a key that indicated how each abbreviation was meant to be expanded.

**University of Minnesota (CASI)**. The CASI dataset contains 37,502 anonymized snippets from the University of Minnesota Medical Center covering 77 unique abbreviations and 410 unique abbreviation–expansion pairs[43,44]. Each snippet is associated with a single abbreviation (for which a span is provided indicating its location in the snippet) and its labeled expansion. We applied a number of filtering steps to the original dataset. First, we only included examples for which the labeled abbreviation–expansion pair was already present in our dictionary, given that we observed poor quality and high noise amongst the remaining pairs. This reduced the dataset to 23,260 snippets covering 64 abbreviations and 123 pairs. We then removed any remaining snippets in which the abbreviation appeared in some modified form in the original snippet (e.g. pm appears as "p.m."), reducing the dataset to 22,822 snippets. Finally, we observed that some snippets in the CASI dataset were quite large, dramatically increasing the time to conduct the sequence alignment necessary for evaluation, and so we removed all snippets that contained greater than 100 tokens, resulting in the final dataset of 21,514 snippets.

We also noticed some discrepancies in the labels of the original dataset that we corrected where possible. For example, we treated the model's expansion as correct in 138 instances where the abbreviation "ra" had the labeled expansion "right atrium" but was expanded by the model to "right atrial". We did so because we observed that the abbreviation appeared before 22 nouns for which such an expansion would be unequivocally correct, including "pressure," "pressures," "mean," "dilation," "function," and "collapse". We also observed 252 instances in which the expansion appeared adjacent to the labeled abbreviation in the original snippet, indicating that the abbreviation itself was used in a complementary fashion and that the phrase would not benefit from further expansion (e.g. "...mri (magnetic resonance imaging) was scheduled..." should not be expanded to "...magnetic resonance imaging (magnetic resonance imaging) was scheduled..."). In these 252 instances, we did not count the model's refusal to expand as a recall failure. There were 118 instances where the abbreviation was "pr," the label "pr interval," and yet the snippet already contained "pr interval," which if correct would have required the model to output "pr interval interval". We excluded these instances from the evaluation. There were also 541 instances in which the labeled abbreviation was used in one of 8 medication brand names: tylenol pm, similac pm, excedrin pm, robitussin dm, mucinex dm, ambien cr, sinimet cr, and paxil cr. If we treat the model's refusal to expand these terms as correct, detection recall increases from 96.8% to 99.1%; however, we did not do so for the main results given that we are evaluating the task of comprehensive abbreviation expansion.

**MIMIC-III**. We used all 59,652 discharge notes from the MIMIC-III dataset[45] to create snippets based on the delimiter of a period followed by a space. We excluded snippets with fewer than 40 characters or greater than 200 characters, and snippets containing bracketed text that indicated certain text was de-identified (de-identification markers text are not in notes seen by patients or physicians). Any remaining duplicate snippets were removed. 95.6% of the 2.4M remaining snippets contained at least one of 3379 expansions from our dictionary. Typical reverse substitution was then performed such that each expansion was replaced with one of its abbreviations at a rate equal to 10,000/(total expansion count) to obtain an expected 10k substitutions for each expansion. If an expansion appeared <10,000 times, all instances of the expansion were replaced. Web-scale reverse substitution was not used since the dataset was much smaller than the web corpus. We then iterated through each snippet and included the snippet in the dataset if any abbreviation within that snippet had not yet been sampled at least two times. This process was similar to the final step in WSRS and was done to ensure that there was no single common abbreviation that overly influenced the metrics. Overall, we generated 6544 snippets containing 3872 unique abbreviation–expansion pairs across 16,872 labeled abbreviations.

**i2b2-2014**. We used the same procedure as described above for MIMIC-III to generate snippets from i2b2-2014 notes[46], with the exception that bracketed text was not excluded, as i2b2-2014 does not use the same deidentification convention. From the 1304 notes in the dataset, we generated 30,878 snippets, of which 91.6% contained at least one of 1675 expansions from our dictionary. Reverse substitution was performed at a rate of 50/(total expansion count) to obtain an expected 50 replacements for each expansion; all instances were replaced if the expansion appeared <50 times. We iteratively sampled snippets according to the same inclusion criteria (snippet contains any abbreviation sampled less than two times so far). The final test set comprised 2888 snippets containing 1913 unique abbreviation–expansion pairs across 5087 labeled abbreviations.

## Assessment of human performance

We took a random sample of 30 snippets from a pool of synthetic snippets that had at least 3 abbreviations and gave them to 3 attending physicians who were board certified in internal medicine, 3 medical students, and 6 lay people who work in health areas. The lay people were all engineers with at least a bachelor's degree who work in a health field to ensure familiarity with using Google search. We asked 3 lay people to "translate" the snippet by expanding all the abbreviations in that sample. These 3 participants were told not to look up abbreviations. We had an additional 3 lay people perform the task with permission to use the Google search engine to try to understand the abbreviations, with a search time limit of 2 min for each snippet. The 6 clinicians (3 attending physicians and 3 medical students) were told not to look up abbreviations to simulate a busy clinical environment. To report the performance of each group we used the average score and calculated 95% confidence intervals using the standard error.

## Abbreviation–expansion dictionary

The abbreviation–expansion dictionary we used for this work combines abbreviation–expansion pairs from a variety of sources:

- Recognition and Disambiguation of Clinical Abbreviations from Vanderbilt University[22] (1202 abbreviations, 2217 abbreviation–expansion pairs)
- Sign-out note abbreviations[54] (448 abbreviations, 765 abbreviation–expansion pairs)
- Beth Israel Deaconess Medical Center abbreviations[59] (1673 abbreviations, 1757 abbreviation–expansion pairs)

- Wikipedia abbreviations (1625 abbreviations, 1890 abbreviation-expansion pairs)
- Additional that were not found in any of these sources added by us (739 abbreviations, 841 abbreviation–expansion pairs)

We deduped these entries and manually reviewed examples based on a priority system, in which we assigned a series of low-quality indicators to each abbreviation–expansion pair, and reviewed those with the most low-quality indicators first. These low-quality indicators were assigned to an expansion if the expansion:

- Contained its own abbreviation
- Contained another abbreviation, although not necessarily its own
- Contained parentheses
- Contained misspellings
- Could not be found in our sampled web corpus

Prioritized manual review resulted in over 550 edits and deletions. The resulting dictionary contains 3758 unique abbreviations and 5794 unique abbreviation–expansion pairs. We make this dictionary available as a part of our data release.

### Computing evaluation metrics

Each dataset had a different source of labels for evaluation. For the web fine-tuning data, MIMIC-III, and i2b2-2014, the long forms in the original text were the labels. For the synthetic snippets, the writer of each snippet generated a key of the intended meaning, which was used as the true label. For the University of Minnesota Dataset, the labels from the dataset were used.

We formulate abbreviation disambiguation as a sequence-to-sequence translation task, meaning that the input snippet is directly translated with abbreviations endogenously detected and expanded, and the exact expansions produced for each abbreviation are not limited to fixed sets of known expansions. To attribute abbreviations in the input sequence to their most likely corresponding expansions in the output sequence, we needed a method for aligning both sequences and extracting these pairs. In cases of an abbreviation surrounded by unmodified text, this extraction process is trivial—we simply anchor on the matched boundaries and extract the pair. However, this process becomes difficult when different expansions appear consecutively with no clear boundaries, which happens often in the clinical text (e.g. "pt has hit complicated by bl civ dvt").

To solve the problem of aligning consecutive expansions, we created a token-level variant of the Needleman–Wunsch global sequence alignment algorithm[47], which is a bioinformatics tool most commonly used to align divergent amino acid (protein) or nucleotide (DNA/RNA) sequences. This dynamic programming algorithm uses a set of scoring rules to determine the optimal alignment between two sequences. Typically, elements from each sequence are aligned, and a score is assigned to each pair of elements, depending on whether the pair are a match, a mismatch, or an insertion/deletion (indel), which happens when an element in one sequence is not paired with anything in the other sequence. An alignment is optimal if it results in an optimal total score derived by summing the scores of all paired elements, so the choice of scoring rules is critical.

We assign match/mismatch/indel scores of 0/1/1, respectively, which results in an optimal total score that equates to the smallest edit distance between the two sequences. In our variation of the algorithm, we also include two custom scoring rules, specifically designed to guide better alignment between an abbreviation and its expansion:

1. Same first character (SFC)—when two mismatched tokens share the first character. Since 91% of the abbreviations in our curated dictionary have corresponding expansions that start with the same character, this scoring rule enables the algorithm to preferentially align abbreviations with the beginning of their corresponding expansion.

2. Expansion gap (EG)—when a gap is introduced by consecutive tokens in the output sequence that are not aligned with tokens in the input sequence, and this gap directly follows an abbreviation in the input sequence and is equal in size to the expected expansion for that abbreviation. This rule was chosen to enable the algorithm to preferentially allow an abbreviation to be matched with multiple expansion tokens following it, rather than cutting the sequence of insertions short. Even if the model uses a valid but incorrect expansion, this scoring rule will allow us to "make room" for many of these attempts, since for 43% of the abbreviations in our dictionary, all of the valid expansions have the same number of tokens.

The match/mismatch/indel/SFC/EG scores we used for our alignments were 0/1/1/0.5/0.6, respectively. We tested various values and evaluated the algorithm's performance by counting the number of pairs the alignment captured that perfectly match the ground truth since these perfect matches are easily detected and counted. We conducted this evaluation on the entire synthetic snippets dataset, with the objective of improving our ability to evaluate performance in an accurate and automated way. This evaluation method did not affect our modeling system in any way.

There were four main measures of performance of the model against these labels: detection recall, detection precision, expansion accuracy, and total accuracy. These are defined in the "Results" subsection "Overview" above. For confidence intervals of the model performance metrics, we generated 400 bootstrap samples of the snippets and reported the 2.5 and 97.5 percentile values alongside the point estimate for each metric.

### Reporting summary

Further information on research design is available in the Nature Portfolio Reporting Summary linked to this article.

## Data availability

The data materials relevant to this study include the following: model parameters data in the form of checkpoints, model fine-tuning data, the dictionary of abbreviations, and external test datasets. Regarding model parameters data, the T5 small, T5 large, and T5 11B pre-trained model checkpoints are publicly available at https://github.com/google-research/text-to-text-transfer-transformer. The 80B checkpoint is unavailable because part of its pre-training data is proprietary. To generate fine-tuning data, we sample snippets from the web containing the long forms of clinical abbreviations. We then substitute these long forms for their abbreviations using web-scale reverse substitution (WSRS), resulting in input-label pairs. We carry out WSRS on the C4 web crawl to generate a dataset called C4-WSRS, which we make available in Tensorflow Datasets (https://www.tensorflow.org/datasets/catalog/c4_wsrs). Instructions for downloading C4-WSRS can be found at https://github.com/google-research/google-research/tree/master/deciphering_clinical_abbreviations. The C4 web crawl itself is publicly available on Tensorflow Datasets as well (https://www.tensorflow.org/datasets/catalog/c4). For our main results, we carry out WSRS on an analogous web crawl with additional filtering similar to that described by Du et al.[42] and medically related filtering as described in the above "Results" section under subsection "Overview". Neither this web crawl nor its WSRS implementation can be released due to their reliance on proprietary code. The manually curated abbreviation–expansion dictionary used for WSRS is available for download at gs://gresearch/deciphering_clinical_abbreviations, which can be easily accessed from the command line with gsutil (https://cloud.google.com/storage/docs/gsutil). There are four test datasets used for evaluation in this paper. The MIMIC-III and i2b2-2014 datasets are available at https://physionet.org/content/mimiciii/1.4/ and https://www.i2b2.org/NLP/DataSets/Main.php respectively.

The code we used to downsample and apply reverse substitution to these datasets, which we describe in detail in the "Methods" section under subsection "Clinical text datasets," is also available on our Github (https://github.com/google-research/google-research/tree/master/deciphering_clinical_abbreviations). CASI is available for download at https://conservancy.umn.edu/handle/11299/137703, and the filtering we applied can be reproduced using details in the "Methods" under subsection "Clinical text datasets" → "University of Minnesota (CASI)". The fourth dataset, which is composed of synthetic snippets covering a range of abbreviations written by clinicians for this study, is available for download at gs://gresearch/deciphering_clinical_abbreviations. We also make two additional resources available at gs://gresearch/deciphering_clinical_abbreviations: the list of 180 clinical abbreviations from our dictionary that we manually identified as English words, and a table of all expansions manually identified as being clinically equivalent for a given abbreviation (e.g. For the abbreviation "abd," the expansions "abdomen exam" and "abdominal exam" were identified as clinically equivalent). These equivalencies are used to treat correct model outputs which do not match the labeled expansion but are clinically equivalent to it.

## Code availability

Data for fine-tuning was from snippets of text collected from the public web that were processed with reverse substitution. We have released the code for C4-WSRS, a fine-tuning dataset generated from the publicly available C4 dataset (described in more detail in the "Results" section under subsection "Data quality") and the code is available on Tensorflow Datasets (https://www.tensorflow.org/datasets/catalog/c4_wsrs). General MC-WSRS pseudocode is provided in Supplementary Algorithm 1. Please refer to the above "Methods" section under the subsection "Fine-tuning dataset generation with WSRS" for more details on the WSRS algorithm. Our training procedure can be exactly replicated using a publicly accessible model training framework such as https://github.com/google-research/t5x (please refer to the "Methods" section under the subsection "Model fine-tuning" above for detailed model training hyperparameters). To evaluate trained models, we make a collection of python libraries publicly available at https://github.com/google-research/google-research/tree/master/deciphering_clinical_abbreviations. The "tokenizer" library contains a custom tokenizer designed for the project, such that abbreviations and expansions are kept as atomic tokens for alignment. The "text_alignment" library contains code to align raw inputs and expanded outputs, as described above in the "Computing evaluation metrics" subsection under "Methods". The "expansion_attribution" library contains code for converting these alignments into the resulting pairs of abbreviations and their expansions, which can then be used for metric computation in the "evaluation" library. This repository also contains a script that can be used to reproduce the evaluation of the T5 11B model outputs. The script downloads the relevant files from gs://gresearch/deciphering_clinical_abbreviations and prints the metrics values to stdout. This repository also contains code to reproduce the reverse-substituted test datasets derived from MIMIC-III and i2b2-2014, using the "text_processing" and "text_sampling" libraries. For more detailed usage instructions, refer to the codebase's README.md file. This code is released under Apache License 2.0.

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

## Acknowledgements

We would like to thank Lisa Williams for her assistance with task formulation, visualizations, and figures. We would like to thank Yun Liu, Arlene Chung, and Andrew M. Dai for the excellent detailed manuscript feedback.

## Author contributions

A.R. and E.L. contributed with study design, statistical analysis, interpretation of results, and drafted and revised the manuscript. A.R., E.L., Y.L., and B.L. created the fine-tuning dataset and the web-scale reverse substitution. Y.L., M.J.C., Y.Z., A.M., and J.G. contributed to machine learning. A.R., E.L., and M.J.C. contributed to the human evaluation. E.L. and J.K. contributed to the calculation of metrics and data analysis. All authors reviewed and edited the manuscript.

## Competing interests

All authors are employed by Google as indicated by the affiliation. Google has filed a provisional patent application 63/269,420 that is related to this article.
