## [Peer Review File · Nature Communications]

Reviewers' Comments:

Reviewer #1:

Remarks to the Author:

This is a very interesting study that is needed in the clinical domain. This study trained large neural language models with novel inference methodologies to decode medical jargon in clinical notes. The proposed approach achieved high accuracy on three test datasets. The authors also compared with laypersons as an additional evaluation.

Major comments

1. The aim of the project is to decode medical jargon for patients or laypersons using neural language models. However, the proposed models just transform the abbreviations in clinical notes to their full forms. These full forms may not be understood by patients. So there is an additional layer that is missing for laypersons to understand medical jargon, i.e., full form of medical jargon to an understandable form of the medical language. This layer of language simplification is related to health literacy. If the aim of the project is to decode medical jargon for physicians, then this may not be an issue. However, the third evaluation on the layperson may become redundant.
2. A lack of experiments on real clinical notes is another major concern in this study. Snippets do not have enough context information: the CASI dataset is a small dataset that doesn't represent the complexity of abbreviations in clinical notes, the MIMIC-III dataset is discharge summaries that are very different from outpatient encounter notes. Again, if the aim of the study is to decode medical jargon for patients (as mentioned in comment 1), these datasets are not suitable as the expansion of abbreviations may not be health literate.

Minor comments

Line 63-64. The statement "there is no pre-existing corpus of medical jargon and more-understandable translations" is not accurate. There are knowledge bases that can be used, for example, UMLS or NCI metathesauruses, and there are also CDC plain language resources and dictionary that contains some relevant information about medical jargon.

Line 133. "...the model must perform multiple expansions in parallel in which successful expansion of one abbreviation requires understanding the potentially ambiguous meaning of another abbreviation." is not clear what the authors try to say.

Line 163. What is the curated dictionary for abbreviation-expansion pairs? Is it publicly available? Figure 1C. It would be great to mention in the figure what dictionary was used to substitute the original word.

Line 183-184. What does "target label" mean? Does it mean the label which indicates the word is an abbreviation or not?

Reviewer #2:

Remarks to the Author:

The paper proposes an approach which identifies potential abbreviations and then expands them. The expansion of an abbreviation also takes into account other abbreviations that are contextually relevant and also ambiguous. Only public, web data is used without the use of a medical domain specific corpus. The technical contributions of the paper include a single-shot approach to abbreviation expansion without the use of medical domain specific data and an inference method. Expansion of medical abbreviations in clinical notes is a challenging natural language understanding task. It requires understanding of context and also the subdomain within the broader medical domain. In the absence of medical jargon specific lexicons, the proposed method can have high significance. The primary strength of the paper is the performance of the proposed model. The mechanism of reverse substitution is a clever approach to generating relevant data without having access to actual clinical notes. However, there are several weaknesses of the paper--low innovation (use of a large transformer-based model is not an innovation from the perspective of this paper). The writing and the comparisons also require substantial improvements. Please see my detailed comments below.

- The evaluation against humans is a thorough strategy and provides good estimates of system

performance.

- Most studies indeed do not use machine learning for identifying clinical abbreviations, but there is some work in that space that was not fully covered in the Related work section.
- In related work, some of the 'recent' works mentioned are actually fairly old considering the speed of current research.
- The strategy used for preparing a large abbreviation dataset by substitution is innovative.
- Transfer learning. The way it is stated in the introduction, it seems as though transfer learning is being heralded as something novel, but it's not. It's a well-explored area in NLP. The use of LLMs also needs to be stated up front. How does this approach compare to other LLMs?
- It is not clear what expertise the physician reviewers had (lines 102-109).
- Ignoring texts that are abbreviations (eg., 'it') -- how does that impact performance?
- Transformer-based model is mentioned in the Results section. There needs to be some description at this point about how this transformer-based model is different or similar to models like BERT or RoBERTa.
- On what basis were the rare abbreviations scaled up while frequent ones were downsampled? At what threshold was a concept considered to be rare? It is unclear from lines 172-177.
- "for MIMIC-III, the label only covers the abbreviations that were synthetically injected" -- does that mean that there were no naturally occurring abbreviations?
- The comparisons showed in the paper are insufficient. There are comparisons against laypersons but not against traditional abbreviation expansion systems (eg., lexicon-based ones). Absence of such baselines makes it impossible to assess how much of an improvement the proposed method provides compared to traditional approaches. Strong traditional approaches should also be included (e.g., the past state-of-the-art approaches that apply machine learning). There are some performance comparisons provided in the Discussion section, but the data are not completely matched. For example, what is the
- "it expands it with an understanding of grammar" -- this is perhaps a slight overstatement of the capability of the model. Contextual disambiguation does not necessarily mean an understanding of grammar.
- The lack of representative clinical note data is mentioned as a limitation. However, there are other openly available clinical datasets that the authors could have used for providing more generalized estimates of their system performance.

Writing is scattered with long and short paragraphs mixed without any comprehensible structure in some places.

Round 1

REVIEWER COMMENTS

Reviewer #1 (Remarks to the Author):

This is a very interesting study that is needed in the clinical domain. This study trained large neural language models with novel inference methodologies to decode medical jargon in clinical notes. The proposed approach achieved high accuracy on three test datasets. The authors also compared with laypersons as an additional evaluation.

Major comments

1. The aim of the project is to decode medical jargon for patients or laypersons using neural language models. However, the proposed models just transform the abbreviations in clinical notes to their full forms. These full forms may not be understood by patients. So there is an additional layer that is missing for laypersons to understand medical jargon, i.e., full form of medical jargon to an understandable form of the medical language. This layer of language simplification is related to health literacy. If the aim of the project is to decode medical jargon for physicians, then this may not be an issue. However, the third evaluation on the layperson may become redundant.

We thank the reviewer for this excellent point. We believe that the journey toward clinical text comprehension begins with disambiguating the text itself. Comprehension of the text involves both recognizing what an abbreviation means and then finding an explanation that makes sense in context of a person's health literacy. One piece of data that we did not expand upon in the initial manuscript is that the human evaluation (Table 2) indicates that laypeople frequently don't understand what abbreviations mean, even with access to Google. This implies that patients couldn't even get started finding online resources to learn more. Having access to an expansion is therefore a key step in helping laypeople understand clinical text. In future work we hope to make the expansions more understandable as well. However, any attempt of making a concept more understandable usually requires simplifying concepts, and the appropriate level of simplification will depend on the knowledge of the particular person - the phrase to describe "EF" (strictly meaning "ejection fraction") would be different to a patient with a physics degree than one with a 5th grade education.

To address the reviewer's specific concerns, we added two early paragraphs in the discussion to help make this clearer:

We found that lay people cannot understand more than half of the abbreviations in clinical text, and even after searching the internet for meaning, they still found 1 in 5 indiscernible. The trained physicians in our study performed substantially better, as expected, but even they were not able to understand all abbreviations, which is consistent with prior research showing physicians routinely read clinical notes that contain unrecognizable abbreviations.²

Our machine learning model can identify thousands of abbreviations across many specialties and expand them with an accuracy that matches or exceeds physicians. High accuracy levels are maintained across 3 independent test sets containing real clinical notes from different health systems, spanning both inpatient and outpatient visits.

We also added a modified paragraph before the conclusion to more directly address the point"

In summary, as patients increasingly gain access to their medical records, decoding the medicelese in notes has the potential to enable better patient understanding by making it possible to search for information about concepts mentioned as otherwise inscrutable abbreviations. Future work is required to solve the last-mile problem⁴⁵ to not only help patients interpret their own medical records in a seamless experience²⁵ but also improve comprehension in those with lower health literacy, which may require decoding and appropriately simplifying text.

2. A lack of experiments on real clinical notes is another major concern in this study. Snippets do not have enough context information: the CASI dataset is a small dataset that doesn't represent the complexity of abbreviations in clinical notes, the MIMIC-III dataset is discharge summaries that are very different from outpatient encounter notes. Again, if the aim of the study is to decode medicelese for patients (as mentioned in comment 1), these datasets are not suitable as the expansion of abbreviations may not be health literate.

We thank the reviewer for the comment, and we would like to respond to the multiple points referenced.

With regards to the concern that "snippets do not have enough context information," we believe our results do not support that assertion. The results on multiple datasets, including the synthetic dataset which contains ground-truth on each abbreviation, show that the vast majority of abbreviations can be disambiguated with the snippets. This is further supported by the fact that expert clinicians were also able to complete the task. We do appreciate the

concern that some abbreviations in clinical practice might need the more patient context but this did not appear to be a major concern given the snippets and datasets we considered.

We respectfully disagree that we lack experiments on real clinical notes. The MIMIC dataset, which is derived all from real notes, contained over 12,000 instances of 2,579 unique abbreviation-expansion pairs. The CASI dataset, which also comes from real clinical notes, contains over 21,000 instances of a smaller number of unique abbreviation-expansion pairs (122) across a much more diverse range of contexts.

We agree with the reviewer that discharge summaries do contain language that may not be seen in typical outpatient encounter notes. Therefore, we have added an additional dataset that explicitly includes outpatient encounter notes (i2b2 - 2014).¹ We included over 4,000 snippets from that dataset covering over 1,380 unique abbreviation-expansion pairs. The results are comparable to the other datasets.

We have therefore demonstrated compelling results over thousands of snippets of real clinical notes from multiple medical centers, spanning inpatient and outpatient notes.

Minor comments

Line 63-64. The statement "there is no pre-existing corpus of medicalese and more-understandable translations" is not accurate. There are knowledge bases that can be used, for example, UMLS or NCI metathesauras, and there are also CDC plain language resources and dictionary that contains some relevant information about medicalese.

We thank the reviewer for this opportunity to clarify the manuscript. Most automated translation systems are trained with actual text in one language along with the same text translated, such as snippets in English along with the same snippets in French. This is different from training on an English to French dictionary because a simple "find-and-replace" strategy using a dictionary does not lead to effective translations of real snippets. We demonstrate the poor performance of this approach in the baseline model in Supplemental Table in the appendix (it has an accuracy of 67% compared to 98% of our language model). Similarly, to train a model to decode medicalese, it's important to have actual text with abbreviations with paired translations and not simply dictionaries of medical terms with their expansions (which the reviewer correctly points out is available in medical knowledge bases such as UMLS and the CDC resources).

We edited the sentence to make our observation more precise:

" For the medical domain, there is no pre-existing corpus of clinical notes with medicalese paired with more-understandable translations, and the words/concepts in the medical text are unique to the domain."

Line 133. "...the model must perform multiple expansions in parallel in which successful expansion of one abbreviation requires understanding the potentially ambiguous meaning of another abbreviation." is not clear what the authors try to say.

We thank the reviewer for pointing out where we can improve the clarity of the manuscript. We have edited the sentence to clarify (we also edited the text around this sentence for improved clarity)

Second, if a snippet contains more than one abbreviation, the model can leverage an expansion of one abbreviation to properly disambiguate another.

Line 163. What is the curated dictionary for abbreviation-expansion pairs? Is it publicly available?

We thank the reviewer for this question. We previously explained the origin of the dictionary in the "description of data for fine-tuning and evaluation" section, but to make it clearer, we have added a new subsection entitled "abbreviation-expansion dictionary" to clarify where the dictionary comes from.

We also plan to publicly release the dictionary with the manuscript - we can provide it to the reviewer or editors ahead of time if needed(our company has an internal review process for data release that is currently underway that we expect to finish soon).

Figure 1C. It would be great to mention in the figure what dictionary was used to substitute the original word.

We thank the reviewer for this excellent suggestion - we have added an explicit mention of the dictionary in the legend for figure 1C.

We take public web pages and identify words or phrases that have corresponding abbreviations (the colored boxes on the left hand side) in the dictionary released along with this manuscript.

Line 183-184. What does "target label" mean? Does it mean the label which indicates the word is an abbreviation or not?

We thank the reviewer for the opportunity to clarify this - we have added an explanation for what target means in the manuscript to clarify it refers to the "ground-truth":

The **ground-truth labels** (i.e. the location and longform/expansion for each abbreviation) for the synthetic snippets were created by the writer of each snippet, whereas the labels for CASI were provided in the original dataset and the labels for MIMIC-III and i2b2-2014 were the sequences before reverse substitution.

Reviewer #2 (Remarks to the Author):

The paper proposes an approach which identifies potential abbreviations and then expands them. The expansion of an abbreviation also takes into account other abbreviations that are contextually relevant and also ambiguous. Only public, web data is used without the use of a medical domain specific corpus. The technical contributions of the paper include a single-shot approach to abbreviation expansion without the use of medical domain specific data and an inference method. Expansion of medical abbreviations in clinical notes is a challenging natural language understanding task. It requires understanding of context and also the subdomain within the broader medical domain. In the absence of medical specific lexicons, the proposed method can have high significance. The primary strength of the paper is the performance of the proposed model. The mechanism of reverse substitution is a clever approach to generating relevant data without having access to actual clinical notes. However, there are several weaknesses of the paper--low innovation (use of a large transformer-based model is not an innovation from the perspective of this paper). The writing and the comparisons also require substantial improvements. Please see my detailed comments below.

We thank the reviewer for examining the manuscript in detail. We would like to address the comment that the "use of a large transformer-based model is not an innovation."

Modeling architecture is only one component of a machine learning system, and recent natural language processing (NLP) research continues to point to the dominance of generalized pretrained "foundation models" on all downstream tasks, regardless of domain. This reflects a growing trend in NLP that has no signs of stopping, and while there is no doubt further model research to be done, this should not discount important research concerning other equally critical components of an end-to-end machine learning system, such as training

data, inference procedures, model explainability and maintenance, etc. If papers that only focus on model innovations are to be published and disseminated, so too should papers that only focus on innovations across these other components, especially when they unlock novel model capabilities or demonstrate solutions to previously intractable problems

We do believe that reformulation of the task as a translation problem *is* novel: prior literature in this domain has focused on a supervised learning approach, mapping abbreviations to a closed set of expansions. A conceptual advance in problem formulation, coupled with concrete evidence that it can solve multiple test sets is novel, in our view.

We welcome suggestions of where the writing could be improved. In addition to specific comments by both reviewers, we have re-drafted the entire discussion section to address this concern and made substantial edits throughout the manuscript..

- The evaluation against humans is a thorough strategy and provides good estimates of system performance.
- Most studies indeed do not use machine learning for identifying clinical abbreviations, but there is some work in that space that was not fully covered in the Related work section.
- In related work, some of the 'recent' works mentioned are actually fairly old considering the speed of current research.

We thank the reviewer for this comment. We have updated the related works section to include newer work, including publications from after this manuscript was originally submitted. We implemented a three-fold methodology to find publications:

- We did a review of pubmed full-text articles published in the last 5 years related to the terms "disambiguating clinical abbreviations"
- We did a similar search in Google Scholar
- We looked at the most recent high-impact publication (Skreta et al in Nature Communications) and evaluated the papers cited by that paper
-

We have selected what we believe are the key related works that we could find, and we would gladly add additional related works that the reviewer suggests.

- The strategy used for preparing a large abbreviation dataset by substitution is innovative.
- Transfer learning. The way it is stated in the introduction, it seems as though transfer learning is being heralded as something novel, but it's not. It's a well-explored area in NLP. The use of LLMs also needs to be stated up front. How does this approach compare to other LLMs?

We thank the reviewer for this comment; we did not intend to convey that transfer learning is novel, but we did intend to convey that traditional forms of transfer learning did not work well in this domain: specifically, training a model on web data and applying it to clinical notes (the simplest form of transfer learning) did not work, as shown in Supplemental Figure 1. We did intend to convey that using iterative and elicitive inference to improve the performance of the model *is* novel. We have re-written the discussion section to hopefully make this point clearer.

The third key challenge we addressed was achieving high performance on clinical notes without using real notes during training: we show that training a language model on a web-scale reverse substitution corpus alone and then applying the model to real clinical notes (transfer learning³⁸) does *not* work well out of the box. We demonstrate this is true for even very large models, suggesting that the web-scale reverse substitution corpus is fundamentally different from clinical text. We improved model performance without using any sensitive clinical data (e.g. few-shot learning or additional fine-tuning) using a novel technique we call elicitive inference, in which the multiple output sequences of a beam search can introduce additional changes. Elicitive inference builds upon a version of iterative inference, in which the model output is fed as input until no changes are introduced,³⁹ and notably achieves a 7 point improvement over it. We highlight that these inference techniques improve model performance without updating any model parameters or using additional input data in the target domain. This work demonstrates that it is possible to protect patient privacy by using public web data to develop systems which achieve expert performance on a clinical notes natural language understanding task.

Regarding the use of large language models, we added an explicit call-out that we use an encoder-decoder transformer model in the first paragraph of the results.

Based on our updated literature review, we have compared our results with the other large-language models evaluated on this task in our discussion:

Recently, Agrawal et al³⁸ demonstrated that large language models can be used in a zero-shot manner to outperform classical disambiguation systems. Our model achieves higher accuracy across

more abbreviations in both CASI (92% on 64 abbreviations vs 90% on 41) and MIMIC (97% on 2000 abbreviations vs 79% on 41), demonstrating that fine-tuning approach can lead to both accurate and scalable results across datasets and abbreviations.

However, our goal was not to compare the advantage of one transformer architecture vs another (we *do* show the effect of model size, which has been shown to be a larger factor than model architecture, all else held equal), so we have added a new paragraph in the discussion to make this clearer.

Second, we do not compare our models to other high-performing language models in the literature, such as decoder-only models.^{35,37,40} Our goal in this work was to leverage T5, a canonical sequence-to-sequence large language model family which has demonstrated state of the art performance on a number of NLP/NLU tasks, and with which we demonstrate expert-level performance on this task. We also assessed how size/capacity within this model family related to performance. Future work could explore comparative performance among different LLMs.

- It is not clear what expertise the physician reviewers had (lines 102-109).

In the results section we have added that the physician reviewers were board certified in internal medicine, and we have also added this in the methods section.

- Ignoring texts that are abbreviations (eg., 'it') -- how does that impact performance?

We thank the reviewer for this question and opportunity to clarify the manuscript. We did not ignore text that are also abbreviations in any of the metrics of the manuscript - all abbreviations were used for evaluations; the specific section titled "abbreviations that are english words" we report the performance on the subset of abbreviations. We have added a sentence to make this clearer in the manuscript.

The metrics above included such abbreviations, but we sought to evaluate whether the model performance was affected in this subset.

Since only 38 of the 890 abbreviation-expansion instances in the synthetic snippets are for abbreviations that are also English words, the performance with or without them would be very similar. We are open to making this explicit in the manuscript, but hope the edit above clarifies the point for most readers.

- Transformer-based model is mentioned in the Results section. There needs to be some description at this point about how this transformer-based model is different or similar to models like BERT or RoBERTa.

We thank the reviewer for the comment. We have tried addressing this point a few ways. First, we explicitly added that we used an "encoder-decoder" T5 model. Since BERT and RoBERTa are encoder-only models, we hope this clarifies the difference. Here is the new text in the results section:

To explore the effect of model size on performance, we quantified the effect of model size by using a set of publicly available transformer encoder-decoder based T5 models with identical pre-training.³⁴

- On what basis were the rare abbreviations scaled up while frequent ones were downsampled? At what threshold was a concept considered to be rare? It is unclear from lines 172-177.

We thank the reviewer for the question. In a loose sense, we used a data-driven approach to define rare based on how often it occurred on the public web. In a stricter sense, we processed the web using distributed computing, and we let each node compute how often it encountered the term in the pages assigned to it. We describe this process in detail in the methods section titled "Fine-tuning with web scale reverse substitution for large language models."

To make this clearer for other readers, we have edited the results section to make this more transparent earlier in the manuscript:

Therefore, we created a more balanced sampling procedure that up-samples rare abbreviations (i.e. the long form was uncommon on the internet),

- "for MIMIC-III, the label only covers the abbreviations that were synthetically injected"
-does that mean that there were no naturally occurring abbreviations?

We thank the reviewer for this comment. We did use natural abbreviations in the synthetic and CASI datasets, but in the original manuscript we previously did not have them for MIMIC. However, based on this comment, we have added an additional analysis on 308 naturally occurring abbreviations for both MIMIC and i2b2, making this the largest set of human evaluation of native abbreviations that we are aware of. The results were consistent with the evaluation performed on abbreviations injected by reverse substitution, which have the advantage of having ground-truth for thousands of different abbreviation-expansion pairs. Here is added text:

We next assessed the expansion accuracy of natively encountered abbreviations (i.e. written by the note author) as opposed to ones introduced by reverse substitution: we randomly sampled 154 unique, native abbreviation-expansions in the MIMIC-III snippets and 154 unique, native ones in i2b2-2014. An attending physician in internal medicine graded each abbreviation-expansion pair in the context of the snippet. In MIMIC-III the expansion accuracy was 95.5% (147/154) and in i2b2 it was 97.4% (150/154), which was consistent with the performance of abbreviations introduced by reverse substitution.

- The comparisons showed in the paper are insufficient. There are comparisons against laypersons but not against traditional abbreviation expansion systems (eg., lexicon-based ones). Absence of such baselines makes it impossible to assess how much of an improvement the proposed method provides compared to traditional approaches. Strong traditional approaches should also be included (e.g., the past state-of-the-art approaches that apply machine learning). There are some performance comparisons provided in the Discussion section, but the data are not completely matched. For example, what is the [sic]

We thank the reviewer for showing us an area where we could clarify some of the work performed and research objectives. First, in addition to a comparison against layperson and human experts, we would like to point out that we do implement a "traditional abbreviation" system in the form of a replacement of each abbreviation with its long form (if multiple long-form existed in the dictionary, we selected the most common one encountered in the open web). We have made it more explicit that this baseline exists in the results section:

We report the raw numbers of all the models in Supplementary Table 1. In that table, we also show the results of a traditional abbreviation expansion system that substitutes abbreviations with its most commonly encountered long-form in the web-corpus; it achieves significantly worse performance than even the smallest language model.

We also would like to identify that we offer multiple very competitive baselines of transformer-based language models on this specific task. We have edited the discussion section to make it clearer what these baselines are intended to show:

Second, we do not compare our models to other high-performing language models in the literature, such as decoder-only models.^{35,37,40} Our goal in this work was to leverage T5, a canonical sequence-to-sequence large language model family which has demonstrated state of the art performance on a number of NLP/NLU tasks, and with which we demonstrate expert-level performance on this task. We also assessed how size/capacity within this model family related to performance. Future work could explore comparative performance among different LLMs.

We do agree in principle with the proposal to use "strong traditional approaches" used by prior investigators. However, the nature of an end-to-end sequence to sequence task is inherently not comparable with the traditional approach of evaluation of a supervised machine learning model to predict among a closed set of possible expansions for an abbreviation that is exogenously provided. Moreover, design choices by prior investigators make the task easier (other papers exclude abbreviations that are also english words, which we do not), so direct comparison is impossible. Finally, we also report performance on 3 benchmark datasets which prior publications have used to help facilitate comparison even given the differences.

To make these issues more transparent, we have re-written the corresponding section in the discussion (including comparison to a more recent manuscript):

We can compare our end-to-end approach with prior state-of-the-art research. Skreta et al trained a model on MIMIC and tested it on a held out test set of MIMIC and also CASI (i.e., an external test set). For the MIMIC internal test set, the investigators report a micro accuracy of 93.5% across 1,116 abbreviations compared to our performance on MIMIC as an external test set (i.e., not trained with MIMIC data) where we achieve 97.0% accuracy across 1,976 abbreviations. On CASI, the investigators report a micro accuracy of 84.1% compared to 92.0% for our model. It is worth noting that a direct comparison between our end-to-end system and previous approaches is not possible, since our system

must identify an abbreviation in addition to expanding it, whereas previous approaches are applied to an exogenously identified abbreviation. This means that for each expansion, our system is evaluated on both identification and expansion, whereas previous approaches are solely evaluated on the expansion. In addition, the abbreviation-expansion pairs evaluated depend on the dictionary used to generate the examples, and the pairs we evaluate our system on are more numerous than those evaluated in previous work.

Recently, Agrawal et al³⁸ demonstrated that large language models can be used in a zero-shot manner to outperform classical disambiguation systems. Our model achieves higher accuracy across more abbreviations in both CASI (92% on 64 abbreviations vs 90% on 41) and MIMIC (97% on 2000 abbreviations vs 79% on 41), demonstrating that fine-tuning can lead to both accurate and scalable results across datasets and abbreviations.

- "it expands it with an understanding of grammar" -- this is perhaps a slight overstatement of the capability of the model. Contextual disambiguation does not necessarily mean an understanding of grammar.

We thank the reviewer for this comment. We agree that the way we worded the text could be interpreted as an overstatement. We therefore have made the sentence clearer:

Importantly, the model can disambiguate the same abbreviation used multiple times with different intended expansions in a single sentence. This demonstrates that each expansion is performed with an implicit understanding of how other abbreviations would be expanded (e.g. "PT" is likely to mean "physical therapy" in the context of "lbp," which means "low back pain").

We do believe the examples do demonstrate the model respects grammar in a non-trivial way, so we have also amended an earlier sentence:

For example "ftm" can be expanded to "trans man" or "female to male" depending on whether it's used as a noun or adjective, demonstrating that the model expands terms in grammatical context.

- The lack of representative clinical note data is mentioned as a limitation. However, there are other openly available clinical datasets that the authors could have used for providing more generalized estimates of their system performance.

We did an inventory of papers cited by the most recent major article on this topic and here are the datasets used in those papers:

	Open Data			Private Data				Non clinical
	UMN/ CASI	MIMIC	i2b2	VUH (private)	Cleveland clinic (private)	Columbia (private)	Mt Sinai (private)	Pubmed
Moon 2012	x							
Xu 2012						x		
Moon 2013	x							
Wu 2015 -	x	x		x				
Joopudi 2018	x				x			
Lui 2018							x	
Li 2019	x	x						
Jin 2019								x
Skreta 2021	x	x	x					
Agrawal 2022	x	x				x		

Based on this inventory, we also evaluated our model on i2b2 data, as described above, which in addition to the two other clinical datasets and synthetic data provide coverage across all previously published publicly available datasets for this task, demonstrating a compelling generalized estimate of model performance.

Writing is scattered with long and short paragraphs mixed without any comprehensible structure in some places.

We have re-written the entire discussion with an eye to make paragraphs more equal in length and we have added many more explicit signposts of the structure in the discussion. We would be open to adding additional sub-titles to improve the structure, if desired.

Reviewers' Comments:

Reviewer #1:

Remarks to the Author:

Thanks for revising the manuscript per reviewers' comments.

1. This reviewer still thinks the paper needs to tune down as the study itself only considered medical abbreviation expansion and disambiguation, not exactly decoding medical language. This reviewer recommends revising the manuscript title.
2. The response to the release of the dictionary for abbreviation-expansion pairs "we can provide it to the reviewer or editors ahead of time if needed(our company has an internal review process for data release that is currently underway that we expect to finish soon)" is not satisfactory. Given that the model and algorithm is not publicly available, this study is not reproducible. Thus it is not a good example of open science.

Reviewer #2:

Remarks to the Author:

Nature comms review

Overall the authors have comprehensively addressed and/or responded to my previous comments. I have several minor comments that are particularly aimed at improving the readability of the paper.

1. Expressions such as Detection recall, expansion accuracy etc. are explained in words. While many readers might prefer that, it maybe a good idea to summarize all of these in the form of mathematical formulations. It would not take up much more space but would—for many readers—improve readability.
2. This is perhaps a choice of writing style and preference, and so I leave it to the authors to decide. Throughout the paper, there are very short paragraphs. In many cases, 1-2 sentence paragraphs that are very similar to preceding and following paragraphs in terms of contents and semantics. Perhaps it would be better to combine such paragraphs into longer ones so that the transitions from one topic subset to another would be better represented by transitions in paragraphs.
3. The comparison with human performance is very insightful.
4. While the proposed system does comparable to medical experts in terms of performance, there is still no guarantee that its outputs are decipherable for lay people.
5. A stronger discussion is required about when the system makes mistakes. Maybe some examples would help and some interpretation as to why the system made specific mistakes. Furthermore, the setting in which such a system is applied may determine the cost of an error. while the number of errors are low, the percentage of errors likely surpasses the percentage of occurrences of many rare expressions. Many such expressions, which can be for example rare medical conditions, may never be accurately expanded by the system. The authors need to discuss this.

Dear Dr. Righetto,

We are pleased to resubmit our revised manuscript "Decoding medical jargon in a privacy protecting machine learning system" In accordance with the valuable suggestions of the reviewers we have revised the manuscript to clarify the points raised. Thank you for your ongoing consideration of our work and for the efforts of the editors and reviewers to improve our manuscript.

As requested, we have included the reviews in full and offer a point-by-point response (in red text).

Sincerely,
Alvin Rajkomar, MD on behalf of all authors

Round 2

Reviewer #1 (Remarks to the Author):

Thanks for revising the manuscript per reviewers' comments.

1. This reviewer still thinks the paper needs to tune down as the study itself only considered medical abbreviation expansion and disambiguation, not exactly decoding medical language. This reviewer recommends revising the manuscript title.

In accordance with the reviewer's thoughtful comment, we have revised the title to "Deciphering clinical abbreviations with a privacy protecting machine learning system."

2. The response to the release of the dictionary for abbreviation-expansion pairs "we can provide it to the reviewer or editors ahead of time if needed(our company has an internal review process for data release that is currently underway that we expect to finish soon)" is not satisfactory. Given that the model and algorithm is not publicly available, this study is not reproducible. Thus it is not a good example of open science.

We thank the reviewer for the comment. We respectfully disagree and believe this study is reproducible. As a part of the study, we release all of the most labor-intensive research materials which pose the largest barrier to reproducibility, such as the nontrivial code required to conduct sequence-to-sequence evaluation, the synthetic snippets that were handwritten for evaluation, the manually labeled clinically equivalent terms, and the hand-curated abbreviation-expansion dictionary. There are elements of the work that rely on proprietary code – mainly web-scale reverse substitution - but we describe in detail the data processing and hyperparameters performed by that code which can be replicated with public web repositories (c4). If there are specific areas of the methods that could be clarified or described in more detail, we would be happy to elaborate.

We would also like to discuss what constitutes good and ethical open science, especially in the field of artificial intelligence. An active area of modern artificial intelligence research is showing how large models can be applied to solve a variety of tasks that were previously not amenable to machine learning, such as complex natural language understanding. Progress is critically measured through performance on key benchmark tasks, which is highly important to understand the capabilities even if the models are not publicly available - a good example is "Language models are few-shot learners" which was published in 2020 and has over 4830 citations despite the largest models not being made publicly available. Knowing that a model can be built with public data and meaningfully perform clinical tasks is important to the field as a general discovery.

Reviewer #2 (Remarks to the Author):

Nature comms review

Overall the authors have comprehensively addressed and/or responded to my previous comments. I have several minor comments that are particularly aimed at improving the readability of the paper.

1. Expressions such as Detection recall, expansion accuracy etc. are explained in words. While many readers might prefer that, it maybe a good idea to summarize all of these in the form of mathematical formulations. It would not take up much more space but would—for many readers—improve readability.

We thank the reviewer for this excellent suggestion. We have added mathematical formulas in the results section to help make the metrics more precise.

2. This is perhaps a choice of writing style and preference, and so I leave it to the authors to decide. Throughout the paper, there are very short paragraphs. In many cases, 1-2 sentence paragraphs that are very similar to preceding and following paragraphs in terms of contents and semantics. Perhaps it would be better to combine such paragraphs into longer ones so that the transitions from one topic subset to another would be better represented by transitions in paragraphs.

We thank the reviewer for this comment. In preparation for the next submission, we have reordered significant sections of the text, shortened aspects, and focused on having more explicit transitions

3. The comparison with human performance is very insightful.

Thank you!

4. While the proposed system does comparable to medical experts in terms of performance, there is still no guarantee that its outputs are decipherable for lay people.

We thank the reviewer for this call out. We hope that renaming the manuscript may help clarify that the current work does not decode text from hard-to-understand concepts. Moreover, we have added an additional limitation to further clarify this.

Fifth, clinical terms, even in their expanded form, can still be unfamiliar to other clinicians or confusing to patients, so accurate expansion does not guarantee clinical comprehension. We acknowledge that additional work is necessary to understand and improve comprehension of clinical notes across a variety of audiences.

5. A stronger discussion is required about when the system makes mistakes. Maybe some examples would help and some interpretation as to why the system made specific mistakes. Furthermore, the setting in which such a system is applied may determine the cost of an error. While the number of errors are low, the percentage of errors likely surpasses the percentage of occurrences of many rare expressions. Many such expressions, which can be for example rare medical conditions, may never be accurately expanded by the system. The authors need to discuss this.

We thank the reviewer for the comment. We have included specific examples of mistakes the model uses in the results section:

"In this evaluation, part of this performance discrepancy was due to the physicians not expanding abbreviations that are commonly used as abbreviations (e.g. not expanding "cm" to "centimeter"); these expansions can be important to non-english speaking patients who rely on translation services which do not work with abbreviations. We show examples of various model and human-made mistakes in Supplementary Table 3"

heent: bilateral cervical and axillary la. notable 1x2 cm node in left axilla which is firm and matted.	Medical Student	head, eyes, ears, nose, throat: bilateral cervical and axillary lymphadenopathy. notable 1x2 cm node in left axilla which is firm and matted.	Here the model did not expand "cm."
	head, eyes, ears, nose, throat: bilateral cervical and axillary lymphadenopathy. notable 1x2 centimeter node in left axilla which is firm and		

model error

We agree with the reviewer that the costs of an error in this domain is unknown, and we have added an additional limitation to make this concern more explicit:

"Finally, the clinical effect of model errors in expanding abbreviations are unknown. Although we demonstrate that the model can expand abbreviations for rare diseases and expressions (Supplemental Table 4), there is no dataset we are aware of that has de-identified notes of a wide sample of rare diseases that can be used to quantify the error rate in these cases. It is unclear whether not attempting to expand rare instances of abbreviations is preferable to expanding them with a given rate of imperfections."